# From Transcription Factors Dysregulation to Malignancy: In Silico Reconstruction of Cancer’s Foundational Drivers—The Eternity Triangle

**DOI:** 10.3390/ijms26209933

**Published:** 2025-10-12

**Authors:** Anna Lisa Cammarota, Albino Carrizzo, Margot De Marco, Nenad Bukvic, Francesco Jacopo Romano, Alessandra Rosati, Massimiliano Chetta

**Affiliations:** 1Department of Medicine, Surgery and Dentistry “Schola Medica Salernitana”, University of Salerno, 84084 Baronissi, Italy; acammarota@unisa.it (A.L.C.); acarrizzo@unisa.it (A.C.); mdemarco@unisa.it (M.D.M.); 2Medical Genetics Section, University Hospital Consortium Corporation Polyclinics of Bari, 70124 Bari, Italy; nenad.bukvic@policlinico.ba.it; 3U.O.C. Oncology A.O.R.N., Cardarelli, 80131 Naples, Italy; francescojacopo.romano@aocardarelli.it; 4U.O.C. Medical and Laboratory Genetics, A.O.R.N., Cardarelli, 80131 Naples, Italy; 5U.O.C. Anatomia Patologica, San Giovanni di Dio e Ruggi d’Aragona, 84131 Salerno, Italy

**Keywords:** transcription factors (TFs), dysregulation of TFs, in silico approach, molecular pathways

## Abstract

Cancer is a multifaceted disease characterized by uncontrolled cell division resulting from substantial disruptions of normal biological processes. Central to its development is cellular transformation, which involves a dynamic sequence of events including chromosomal translocations, genetic mutations, abnormal DNA methylation, post-translational protein modifications, and other genetic and epigenetic alterations. These changes compromise physiological regulatory mechanisms and contribute to accelerated tumor growth. A critical factor in this process is the dysregulation of transcription factors (TFs) which regulate gene expression and DNA transcription. Dysregulation of TFs initiates a cascade of biochemical events, such as abnormal DNA replication, that further enhance cell proliferation and increase genomic instability. This microenvironment not only sustains tumor growth but also promotes the accumulation of somatic mutations, thereby fueling tumor evolution and heterogeneity. In this study, we employed an in silico approach to identify TFs regulating 622 key genes whose mutations are implicated in carcinogenesis. Transcriptional regulatory networks were analyzed through bioinformatics methods to elucidate molecular pathways involved in cancer development. A thorough understanding of these processes may help to clarify the function of dysregulated TFs and facilitate the development of novel therapeutic approaches designed to make cancer treatments personalized and efficacious.

## 1. Introduction

Spatial and temporal variations in the metabolic pathways that regulate cellular function significantly influence the complex interactions between mutational processes and epigenetic alterations, resulting in the formation of tumor genomes. The accumulation of mutations is affected by these mechanisms, which lead to the dysregulation of various types of regulatory elements. These alterations promote the establishment of a tumorigenic state by fostering the disruption of normal cellular homeostasis. The functional and genetic mechanisms that underpin these processes remain predominantly elusive, notwithstanding advancements in cancer research [1].

Despite notable advances in cancer research, the functional and genetic mechanisms underlying these processes remain largely unresolved. Within this context, transcription factors (TFs) play a pivotal role. By binding to specific DNA sequences, TFs act as master regulators of gene expression and provide the basis for the conceptual framework we define as the “Eternity Triangle.” This framework is not intended as a metaphor but represents the central theoretical innovation of this work. It integrates three fundamental regulatory axes: proliferative renewal, differentiation, and the controlled management of mutational processes. In physiological conditions, these axes operate in concert to sustain cellular homeostasis, enabling tissues to regenerate, maintain functional integrity, and preserve genomic stability through accurate DNA repair and, when appropriate, regulated mutagenesis such as that observed in immune diversification.

When TF activity is balanced, this triangular network maintains a delicate equilibrium across these programs. However, dysregulation of TFs disrupts this equilibrium: proliferative control is bypassed, differentiation pathways are impaired, and regulation of mutational load collapses, leading to catastrophic genomic instability. The breakdown of this system generates the oncogenic triad of uncontrolled proliferation, loss of cellular identity, and the emergence of immortalized cells, which together capture the essence of malignant transformation.

Such conditions foster an environment that promotes excessive mutation accumulation, genomic instability, and impaired differentiation. Consequently, TF dysregulation emerges as a central driver in the initiation, progression, and therapeutic resistance of cancer, as well as a key pathogenic mechanism in other degenerative and proliferative diseases. Replication stress, characterized by decreased fidelity during DNA replication, serves as a critical mechanism by which TFs facilitate carcinogenesis. This phenomenon can result in genomic instability, stalling or collapse of replication forks, and the accumulation of somatic mutations that stimulate tumor progression [2]. MYC (MYC Proto-Oncogene) and STAT5 (Signal Transducer And Activator Of Transcription 5A) are two examples of TFs that are directly related to replication stress and genomic instability [3]. MYC is an essential modulator of apoptosis, cellular transformation, and the cell cycle. Solid tumors and hematological cancers are among the many cancers linked to its overexpression or mutation. By overloading the replication machinery and raising the possibility of replication mistakes, MYC accelerates the cell cycle, while STAT5 plays a role in cell survival and proliferation [4,5].

TFs also play a critical role in DNA repair mechanisms, overseeing important pathways such as homologous recombination (HR), nucleotide excision repair (NER), and mismatch repair (MMR) [6,7]. A prominent example is the DNA-binding TF p53 (Tumor Protein P53), which activates genes involved in DNA damage repair and cell cycle arrest. Tumorigenesis accelerates when p53 is altered or inactivated, as cells accumulate mutations, become unable to repair genetic damage, and evade apoptotic processes [8]. Similarly, other TFs that regulate genes like *BRCA1/2* (BRCA1/2 DNA Repair Associated), essential for homologous recombination, may lead to genomic instability if dysregulated, increasing the risk of mutations and tumor progression [9].

TFs are also essential for cell cycle regulation because they influence the expression of genes that code for phase-transition-related proteins. The G1-to-S phase transition, for example, depends on the production of cyclins and cyclin-dependent kinases (CDKs), which are regulated by E2F (E2F Transcription Factor) [10]. Cell cycle progression is disrupted by dysregulation of E2F, which accelerates proliferation and leads to the accumulation of mutations [11]. The tumor suppressor protein RB (retinoblastoma), which is controlled by E2F and stops cell division when DNA damage is present, is another example. Bypassing this regulation, E2F dysregulation permits damaged cells to proliferate rapidly [12].

TFs are also involved in signaling pathways that control cell division and proliferation. Hyperactivation of tumor-promoting networks results from changes in these pathways caused by mutations in TFs or their upstream regulators. For instance, dysregulation of the mitogen-activated protein kinase (MAPK) pathway’s ELK1 (ETS Transcription Factor ELK1) and ETS (ETS Proto-Oncogene) leads to an increase in the expression of genes that promote cellular proliferation [13]. Similarly, these pathways may become persistently hyperactivated due to alterations in upstream proteins such as the Ras/Raf/MAPK pathway [14]. Another crucial example is the PI3K/AKT/mTOR pathway. When AKT (a serine/threonine kinase) is hyperactivated, it promotes uncontrolled cell proliferation and survival even in unfavorable conditions. AKT achieves this function by acting on several TFs, such as FOXO (Forkhead Box O). Normally, FOXO negatively regulates genes involved in cell cycle arrest and apoptosis. However, when AKT is overactive, it inhibits FOXO, decreasing its ability to promote these functions and thereby reducing cell cycle arrest and apoptosis [15].

TFs also are involved in R-loop formation near TSS (Transcription starting site) by attaching to specific DNA spots. R-loops are special structures made of RNA and DNA that appear when genes are being copied. If R-loops are not properly controlled, often because TFs are not working right, they can cause damage to the DNA and problems during cell division. This correlation between transcriptional activity and mutational susceptibility emphasizes how crucial gene regulation is to the development of tumors [16]. Moreover, to meet the metabolic demands of cancer cells, high transcriptional activity exposes DNA to transcriptional and replicative stress [17]. As a consequence, damaged DNA not only directly causes mutations but also increases the likelihood of further DNA damage and mutations. This cascading effect leads to altered gene expression and faster tumor growth [18]. Furthermore, actively transcribed genes’ TSSs are found in less compact, open chromatin domains, which make TF’s access easier [19]. Despite being essential for transcriptional activation, this arrangement renders the TSS more susceptible to damage from endogenous sources (such as reactive oxygen species, ROS) and exogenous sources (including UV radiation, mutagenic chemicals, and viruses) [20]. It is suggested that highly transcribed genes are also the most prone to mutations, as such damage often manifests as somatic mutations that accumulate at TSS, with incidence proportional to mRNA abundance [21].

Indeed, mutations are abundant in genomic regions of open chromatin, demonstrating the close relationship between transcriptional activity, chromatin state, and DNA replication in regulating the distribution of mutations [22]. When the genome is analyzed at intermediate resolutions (hundreds or thousands of nucleotides) or megabases, unique mutational patterns linked to different functional genomic regions are revealed. The highest mutation frequency in cancer genomes is seen in exons, transcription factor binding sites (TFBSs), and chromatin architectural elements [23]. For example, active gene promoters, such as those involved in melanoma, tend to accumulate UV-induced C>T somatic mutations because DNA-binding regulatory proteins modulate nucleotide excision repair efficiency in these regions, leading to localized variability in repair and mutation patterns [24]. Additional players in this context, the architectural protein CTCF (CCCTC-binding factor) and the binding sites of key transcriptional regulators are frequently subject to somatic mutations across various cancers; besides their roles in transcriptional regulation, they are essential for maintaining chromatin accessibility and genomic stability by organizing the three-dimensional genome architecture [25,26].

Of particular significance are mutations in the promoter of the *TERT* (telomerase reverse transcriptase) gene. These mutations create binding sites for ETS family TFs, leading to constitutive activation of *TERT* [27]. Aberrant *TERT* expression enables cancer cells to acquire replicative immortality, a hallmark of tumor progression. *TERT* activation also confers a selective advantage to tumor cells by promoting proliferation, migration, and invasiveness, key elements of metastasis [28]. Although such promoter mutations are relatively rare, their role is particularly significant in aggressive cancers such as melanoma, glioblastoma, and urothelial carcinoma [29].

The present study employed an in silico approach to identify TFs involved in the regulation of 622 key genes associated with carcinogenesis when mutated. Transcriptional regulatory networks were analyzed using bioinformatics methodologies to describe interconnected molecular pathways implicated in cancer development. This integrative computational analysis investigated TFs that potentially contribute to increased mutation rates, accelerated cellular proliferation, and alterations in transcriptional and signaling networks, ultimately leading to heightened genomic instability. The approach enabled mapping of molecular networks regulated by TFs and the identification of specific factors whose dysregulation could represent common drivers across various cancer types. These findings provided insights into mechanisms underlying tumorigenesis and suggested potential involvement of novel molecular pathways in the progression of specific cancer subtypes.

## 2. Results

### 2.1. TF Families Identified as Interactors in Oncogene Promoters

A fundamental question in cancer genomics is whether the promoters of oncogenes share common regulatory architectures that reflect their roles in tumorigenesis. To address this, we developed a comprehensive bioinformatics pipeline to systematically identify transcription factor binding sites (TFBS) enriched across a curated set of 622 cancer-driving genes from IntOGen (v2024.09.20). Our analysis revealed a striking convergence in transcriptional regulation. We discovered five distinct DNA motifs significantly overrepresented in the promoters of these oncogenes. Computational homology mapping against the HOCOMOCO v11 database linked these motifs to a network of 128 transcription factors (TFs), implicating them as master regulators of carcinogenesis. This finding suggests that despite the vast genetic heterogeneity of cancer, a core set of transcriptional programs is recurrently co-opted to drive oncogenic expression [30].

The identified TFs are not isolated DNA-binding proteins but rather key nodes within the regulatory circuitry governing cancer hallmarks. They are significantly enriched for functions related to cell proliferation, apoptosis evasion, metastasis, immune modulation, DNA damage response, and tumor progression. The evolutionary conservation of their DNA-binding domains highlights the critical and non-redundant nature of these interactions.

Our de novo motif discovery approach, performed with MEME-ChIP, enabled the identification of a high-confidence set of TF–promoter interactions that extend beyond known associations. This robust and unbiased methodology offers a comprehensive view of the transcriptional landscape co-opted in cancer and provides a valuable resource for elucidating oncogenic mechanisms and identifying potential therapeutic targets. Importantly, the transcription factor families presented in Table 1 do not represent a literature-derived enumeration; they are the curated output of our in silico MEME-ChIP/Tomtom pipeline. This dataset constitutes the analytical foundation of our study, whose significance becomes evident when integrated with the subsequent interaction network (Section 2.3). By transforming motif discovery into a systems-level perspective, the analysis moves beyond the scope of a review article and underscores the novelty of our contribution.

The evaluation of TFs revealed several important protein families that are essential for controlling gene expression and the molecular mechanisms behind carcinogenesis (Table 1).

E2F Family

The E2F family (E2F1, E2F3, E2F4, E2F6, E2F7) plays a crucial role in the transition of the cell cycle from G1 to S phase by controlling genes essential for cell cycle progression. Mutations or post-transcriptional changes that disrupt these regulators lead to unchecked cell division, which contributes to malignancies including retinoblastoma, breast cancer, and other carcinomas [31].

MYC Family

Another key oncogenic member is from the MYC family: the MYCN oncogene (MYCN Proto-Oncogene, BHLH Transcription Factor), which plays a critical role in neuroblastoma, and its amplification is associated with a poor prognosis [32]. As described in neuroblastoma, but also in lung carcinoma, lymphoma, and breast cancer, MYC hyperactivation can enhance tumor cell survival and proliferation. Moreover, MYC contributes to tumor growth by interacting with other cellular signaling pathways, including the PI3K/AKT pathway and cyclin-dependent kinases (CDKs) [33].

KLF Family

Members of the KLF (Kruppel-like factors) family, including KLF1, KLF3, KLF6, KLF12, and KLF15, have dual functions [34]. For instance, KLF6 functions as a tumor suppressor, for instance, controlling metastasis, invasiveness, and cell proliferation. In prostate, liver, and colon cancers, its expression is often reduced or missing [35]. However, in other cancers, KLF3 and KLF15 might have carcinogenic properties. For example, KLF15, promotes uncontrolled growth and resistance to treatment in malignancies of the liver and prostate. Additionally, the oxidative stress response, cellular senescence, and angiogenesis are all impacted by the KLF family. The balance between apoptosis and proliferation is disrupted when these factors are not working properly, which increases the development of tumors [36].

FOXO Family

The regulation of cell proliferation, survival, and apoptosis also relies on the FOXO family (Forkhead Box O), which comprises FOXO1, FOXO3, and FOXO4 [37]. As tumor suppressors, FOXO factors control the expression of genes associated with apoptosis, DNA repair, and stress responses. In cancers, the loss of FOXO function is prevalent, as tumor cells frequently evade programmed cell death by deactivating FOXO through mechanisms such as phosphorylation [38]. By inhibiting FOXO’s nuclear translocation, processes like phosphorylation by kinases such as AKT (AKT Serine/Threonine Kinase) diminish its ability to activate pro-apoptotic target genes. The loss of FOXO family members heightens invasiveness, chemoresistance, and metastasis, and is linked to malignancies of the skin, liver, lung, and prostate. Restoring FOXO3′s activity may offer promising therapeutic strategies for tumors that are resistant to treatment, as it has been recognized as a vital TF in anticancer responses [39].

SP Family

The SP family, which includes SP1 (Sp1 Transcription Factor) and SP3 (Sp3 Transcription Factor), is critical for controlling biological functions such as cell cycle progression, DNA repair, and stress response. SP1 regulates many genes required for cell proliferation and survival, including those that control the cell cycle and DNA synthesis [40]. Overexpression of SP1 is frequently associated with a poor prognosis in malignancies such as breast carcinoma, as it enhances tumor development, invasiveness, and chemoresistance [41].

GATA Family

The GATA family (GATA Binding Protein), which includes GATA2, GATA3, and GATA6, governs critical cellular processes such as cell migration, epithelial-to-mesenchymal transition (EMT), and metastasis, all of which play important roles in cancer formation and progression [42]. In breast cancer, for example, GATA3 expression has been described as an adjunctive marker linked to a poorer overall survival in triple-positive breast cancer patients [43]. GATA6 plays a context-dependent role in lung adenocarcinoma progression by modulating chromatin accessibility and transcriptional programs, thereby influencing tumor cell proliferation and differentiation based on the cell type and genetic background [44]. GATA2, which regulates immunological responses and stem cell maintenance, has also been recently linked to chemotherapy resistance in acute myeloid leukemia by regulating RASSF4 to suppress p53-mediated apoptosis [45].

STAT Family

Migration, invasion, and cell survival-related gene expression are regulated by the STAT family (Signal Transducer and Activator of Transcription), particularly STAT3. By stimulating genes related to cell proliferation, apoptosis resistance, and metastasis, STAT3 is frequently active in malignancies such breast cancer, lung carcinoma, and melanoma, which accelerates the course of the disease [46]. STAT3 also promotes tumor progression and immune evasion by driving tumor cell resistance to natural killer (NK) cell-mediated killing while also regulating NK cell functions, making it a critical target for improving NK cell-based cancer immunotherapies [47].

IRF Family

Apoptosis and immunological responses are controlled by the IRF (Interferon Regulatory Factors) family and members such as IRF1 and IRF8 are crucial for anti-tumor immunity because they activate T cells and dendritic cells, promoting the release of pro-inflammatory cytokines [48]. Their essential role in tumor-immune system interactions is underscored by the fact that altered IRF activity can lead to immune suppression and tumor progression [49].

NF-κB family

Another family of TFs commonly linked to carcinogenesis is the NF-κB family (Nuclear Factor kappa-light-chain-enhancer of activated B cells). It is well established that NF-κB1 (NFAC1) and other members of the NF-κB family are activated in various malignancies, contributing to a persistent inflammatory environment and resistance to apoptosis [50]. The production of pro-inflammatory cytokines, chemokines, and other substances that promote inflammation—a crucial component in the growth and evolution of tumors—is stimulated by NF-κB activation [51]. Chronic inflammation induced by NF-κB in many malignancies fosters a milieu that contributes to neoangiogenesis, cell division, and metastasis [52]. Furthermore, by triggering the expression of molecules that inhibit apoptotic signals, NF-κB significantly contributes to resistance to chemotherapy and radiation therapy, allowing cancer cells to withstand treatment. To reduce tumor growth and enhance treatment efficacy, therapies that disrupt this inflammatory and anti-apoptotic signaling pathway can target the hyperactivation of NF-κB, a characteristic of many cancers, including skin, colon, and breast cancers [53].

SOX and HOX Family

The pluripotency of cancer stem cells, crucial for tumor persistence and recurrence, is strongly linked to the SOX2 (SRY-Box Transcription Factor 2) and HOXA9 (Homeobox A9) TF families [54,55]. Specifically, SOX2 is a TF that controls the differentiation and pluripotency of stem cells and has been linked to the development of several aggressive malignancies, including lung carcinoma and glioblastoma. By preserving a reservoir of cancer stem cells that contribute to treatment resistance and recurrence, SOX2 promotes tumor growth and the possibility of metastasis in glioblastoma [56]. Likewise, in lung cancer, SOX2 has a role in controlling cellular plasticity and the tumor cells’ capacity to adjust to changes in their surroundings, which helps them survive even under harsh circumstances [56]. Maintaining pluripotency and controlling stem cell differentiation are additional functions of the HOXA9 family, which is a component of the larger Hox complex. In leukemia and solid tumors, HOXA9 has been specifically linked to the invasiveness and proliferation of cancer cells, which promotes the advancement of the disease and resistance to treatment [57].

SMAD Family

The cell cycle, DNA repair, and cell survival are fundamentally regulated by the SMAD family members, which are driven by TGF-β (Transforming Growth Factor-beta) signaling and linked to the activation of several processes such as apoptosis, cellular senescence, and differentiation requires SMAD-mediated signaling [58]. Many malignancies are characterized by genomic instability, which can result from SMAD protein dysfunction caused by either mutations or altered activation signals. In pancreatic cancer for example, loss of SMAD4 impairs T-cell infiltration and chemokine production, leading to a poorly immunogenic tumor microenvironment with reduced IFNγ-driven PD-L1 expression [59].

Sex hormone receptor families

Sex hormones, through their receptors, such as the androgen receptor (AR) and estrogen receptors (ERα and ERβ), play crucial roles in the development, progression, and treatment resistance of hormone-responsive cancers like prostate, breast, ovarian, and endometrial cancers. Deregulated AR and ER signaling contributes to tumor growth and complexity, making these pathways key targets for developing more effective therapies against these cancers [60]. Prostate cancer, the most common cancer in men, progresses from an androgen-dependent to a castration-resistant state driven by androgen receptor (AR) alterations, leading to treatment resistance often enhanced by epigenetic changes and tumor microenvironment interactions [61]. Similarly, alterations in the estrogen receptor alpha gene (*ESR1*), such as amplification, mutations, and rearrangements, drive therapeutic resistance and metastasis in estrogen receptor positive (ER+) breast cancer. Understanding these ESR1 changes offers potential avenues for developing treatments to improve outcomes in patients with advanced, metastatic disease [62].

ETS Family

Within the ETS (ETS Proto-Oncogene) family, numerous members are crucial for regulating tumor cell proliferation, migration, and invasiveness; for example, ETS1 and ETS2, are known for their influence on tumor growth and invasiveness thereby regulating expression of genes that control the cell cycle, metastasis, and neoangiogenesis, promoting tumor progression [63]. Moreover, ETS-1 also influences the tumor immune microenvironment by regulating immune cell functions and interacting with other signaling molecules in several malignancies, making it an important target for cancer diagnosis, prognosis, and therapy [64].

ZNF Family

The ZNF (Zinc Finger) family includes TFs such as ZNF18 and ZNF250, which are involved in gene regulation and modulation of tumor cell behavior [65]. Recently, the overexpression of the ZNF560 protein in osteosarcoma has been associated with poor patient prognosis, while its downregulation reduces tumor cell viability and migration and induces apoptosis, indicating its potential as a predictive biomarker for the disease [66].

PRDM Family

Another relevant group of TFs in tumorigenesis includes PRDM1 and PRDM6 (PR/SET Domain 1 and PR/SET Domain 6) involved in gene repression and epigenetic regulation; their alteration is associated with numerous mechanisms of therapeutic resistance and vascular proliferation regulation [67]. PRDM1 overexpression in hepatocellular carcinoma has been described as promoting PD-L1 expression via the USP22-SPI1 pathway, leading to CD8+ T cell exhaustion and immune evasion, while combination treatment with PD-1 blockade enhances anti-tumor immunity, suggesting a novel therapeutic strategy for HCC [68]. PRDM6 has been described as having the ability to alter chromatin structure and gene expression in human neuroepithelial stem cells, hence increasing the development of medulloblastoma. It has oncogenic potential, and blocking PRDM6 may represent a treatment option for cancers that express it [69].

Retinoic receptor Family

Cellular differentiation, proliferation, and death are all critically regulated by retinoic acid receptors, including retinoic acid receptor alpha (RARA) and retinoic acid receptor gamma (RARG) by interacting with these receptors, retinoic acid modifies the expression of genes related to cell survival and differentiation [70]. In acute promyelocytic leukemia (APL), retinoid receptor function is disrupted, demonstrated by the inhibition of RARA caused by a chromosomal translocation that creates the PML/RARA fusion protein, which blocks the differentiation of myeloid cells and drives cancer cell proliferation [71]. Dysregulation of RARA and RARG has also been linked to cell survival and chemoresistance in other solid tumors, including skin and lung cancers. The potential of retinoid modulators as treatments for various malignancies associated with retinoic receptor abnormalities is currently being explored. For example, retinoic acid therapy, which aims to restore receptor function, has shown promising results in the treatment of APL [72].

EGR Family

Cellular reactions to external stimuli like stress, growth hormones, and inflammatory signals depend on TFs belonging to the Early Growth Response (EGR) family, which includes EGR1 and EGR2. These elements control several cellular functions, including immunological responses, differentiation, and proliferation [73]. Evidence from various malignancies shows that EGR1 can function as a tumor suppressor by monitoring DNA damage, promoting tumor cell apoptosis, and enhancing the effectiveness of radiotherapy and chemotherapy. Conversely, in specific tumor microenvironments like hypoxia, elevated EGR1 expression supports tumor cell survival, proliferation, metastasis, and neoangiogenesis [74].

MEF2 Family

The development and survival of tumor cells depend on the involvement of the MEF2 (Myocyte Enhancer Factor 2) family, which includes MEF2A, MEF2B, and MEF2D; these TFs are involved in signaling pathways that regulate cellular homeostasis and metabolic adaptation [75]. Although MEF2A and MEF2D have been linked to differentiation of muscle and cardiac cells, in the absence of regular growth signals, tumor cells may proliferate due to dysregulation of the MEF2 family, which can cause unchecked growth and aberrant metabolic adaption [76]. For instance, abnormal MEF2D activity is associated with treatment resistance, invasiveness, and tumor progression in some cardiac and muscular sarcomas. MEF2 may also control the synthesis of chemicals that support neoangiogenesis in solid tumors, which would help the tumor grow by supplying the oxygen and nutrients it needs. Therefore, blocking MEF2 signaling pathways is thought to be a viable therapeutic approach to reduce tumor growth and metastasis [77]. Table 2.

### 2.2. TFs Associated Signaling Pathways and Their Role in Tumorigenesis

Building upon these findings, the list of TFs identified through motif analysis was subsequently analyzed using STRING, a platform designed to explore both physical and functional interactions among proteins. This step enabled the construction of a comprehensive interaction network, revealing a dense web of connections among the candidate TFs. Many of these interactions were supported by high confidence scores, indicating strong evidence for their biological relevance. The pathways analysis highlighted several signaling pathways that coordinate vital cellular functions required for tumorigenesis, cancer growth, and advancement that are intimately associated with TFs activity [78] (Figure 1).

The WNT signaling pathway is particularly significant due to its activation in early carcinogenesis and its crucial role in cancer stem cells in tumors such as breast and colon cancer [79]. In addition to encouraging self-renewal, this pathway affects cell polarity and adhesion, both of which are essential for tissue integrity. Dysregulation of the WNT pathway results in β-catenin accumulation, promoting transcription of genes involved in proliferation and survival, thereby fostering tumor-supportive microenvironments [80]. Additionally, WNT signaling influences how tumor cells interact with their surroundings, which promotes immune evasion and cellular plasticity, both of which accelerate the spread of the tumor. The pathway’s pivotal role in the development of aggressive, therapy-resistant tumor states is highlighted by these synergistic effects [81].

DNA damage and cellular stress trigger the activation of the p53 tumor suppressor pathway, which is essential for preserving cellular integrity. The p53 pathway destroys damaged cells and stops the spread of mutations that could lead to malignant transformation through processes such DNA repair, cell cycle arrest, and apoptosis [82]. Loss or malfunction of p53 impairs the cell’s ability to repair genetic damage, leading to an accumulation of mutations that may drive malignancy. Additionally, by preventing treatment-induced apoptosis, p53 inactivation contributes to anticancer therapy resistance [83]. Mutations or amplifications that lead to the overexpression of the *MYC* gene cause normal cells to transform into aggressive, highly proliferative cancer cells [84]. The MYC pathway, which primarily operates in the nucleus of rapidly proliferating cells, activates genes that are essential for DNA replication, protein synthesis, and cell survival. Dysregulation of *MYC* not only drives tumor growth but also promotes angiogenesis and immune evasion, thereby altering the tumor microenvironment. Although direct targeting of *MYC* remains challenging due to its complex molecular interactions, its critical role in cancer cell dependency makes it an attractive therapeutic target [85]. The Notch signaling pathway is a multifunctional regulator of tumor development, influencing growth, metastasis, and cellular plasticity. This pathway operates through paracrine or autocrine signaling involving interactions between Notch ligands and receptors on the cell surface, culminating in the translocation of the Notch intracellular domain to the nucleus to regulate gene expression [86]. Aberrant Notch activation promotes cancer cell survival and angiogenesis, which are crucial for tumor growth and nutrient supply. The pathway’s role in epithelial-to-mesenchymal transition (EMT) is critical for invasiveness and metastasis, facilitating cancer cells acquiring enhanced motility and resistance to therapy. Notch signaling exhibits both oncogenic and tumor-suppressive effects depending on the tissue context, highlighting its biological complexity [87]. The NF-κB pathway has a central role in this context; it serves as a key link between chronic inflammation and tumorigenesis. Activated by inflammatory stimuli such as cytokines, growth factors, and infectious agents, this pathway initiates a signaling cascade leading to the nuclear translocation of the NF-κB factors complex [88]. In the nucleus, it orchestrates the expression of genes driving cell proliferation, survival, and angiogenesis, while simultaneously promoting an immunosuppressive tumor microenvironment. In inflammation-associated cancers like hepatocellular carcinoma and gastrointestinal tumors, NF-κB promotes tumorigenesis by facilitating the accumulation of mutations through reactive oxygen species (ROS) production and recruiting tumor-supportive immune cells. Additionally, its activation contributes to therapy resistance by perpetuating the cancer-inflammation feedback loop [89]. The Hippo pathway is a highly conserved molecular network crucial for controlling cell proliferation and preventing excessive growth. Early activation of the Hippo pathway maintains a balance between cell proliferation and differentiation, thereby regulating organ size and preventing hyperplasia [90]. In cancers, however, dysregulation of Hippo signaling, often through the loss of kinases MST1/2 and LATS1/2, activates YAP/TAZ coactivators. These coactivators translocate to the nucleus, driving the expression of genes associated with cell growth, apoptosis resistance, and EMT, thereby promoting invasiveness and metastasis. Hippo signaling is particularly active in the nucleus and cytoplasm of cancer stem cells, supporting sustained proliferation and therapy resistance [91]. Alterations in PI3K, PTEN, or receptor tyrosine kinases often deregulate the PI3K/AKT pathway, a key molecular driver of cancer cell survival and growth [92]. Extracellular growth signals fuel numerous cellular activities, which are then carried out by activated AKT, including metabolic reprogramming to maximize nutrition absorption and energy efficiency, cellular growth support through mTOR activation, and apoptosis suppression (e.g., via BAD phosphorylation) [93]. While the PI3K/AKT pathway primarily functions in the membrane and cytoplasm, it also influences the nucleus, where it regulates key genes involved in tumor development. Chemotherapy and targeted therapy resistance are frequently linked to this pathway hyperactivation [94]. Growth factors also activate the MAPK signaling pathway, a kinase cascade involving RAF, MEK, and ERK; this pathway regulates oxidative stress responses, differentiation, and proliferation. Uncontrolled signaling and increased carcinogenesis result from oncogenic mutations in elements such as *BRAF* or *KRAS* [95]. The TGF-β signaling pathway contributes to the development of cancer in two ways. By preventing cell division, it suppresses tumors in their early stages. However, in later stages, TGF-β switches to promoting tumor growth [96]. TGF-β promotes tissue invasion and metastasis by triggering EMT [97]. It interacts with the tumor microenvironment, affecting immune cells, fibroblasts, and the extracellular matrix in addition to epithelial tumor cells. Additionally, TGF-β suppresses antitumor immune responses and creates a protective niche for tumors by modulating immune evasion [98]. The adaptation of cancer cells to the low oxygen levels found in rapidly expanding solid tumors depends on the hypoxia-induced pathway. Hypoxia stabilizes HIF-1α, which in turn activates genes that control angiogenesis (e.g., VEGF) and metabolic adaptations for survival in harsh environments [99]. In addition to maintaining tumor growth and promoting the development of new blood vessels, this pathway also plays a role in treatment resistance and the spread of cancer. In addition to increasing flexibility and encouraging EMT and cancer stem cell survival, hypoxia also favors more aggressive tumor cell clones [100,101]. Beyond classical caspase-mediated pathways, novel therapeutic strategies like oncolytic peptides can induce apoptosis through membrane disruption and activation of intrinsic pathways involving Bax/Bcl-2 balance, showcasing a promising synergistic oncolytic-immunotherapy effect This equilibrium is frequently upset in tumors, which results in increased cell survival and suppression of apoptosis [102]. One important regulator of cellular differentiation, apoptosis, and tumor suppression is the retinoic acid pathway, which is regulated by retinoic acid receptors (RARs and RXRs) [103]. Compounds like all-trans retinoic acid (ATRA), which causes young leukemic cells to grow into less proliferative phenotypes, are used therapeutically to exploit this pathway in hematologic malignancies like acute promyelocytic leukemia (APL) [104]. Mutations or decreased receptor expression, however, may cause the system to malfunction in different cancer types. In addition to controlling immunological and epithelial cell differentiation, retinoic acid also affects the tumor microenvironment by modifying angiogenesis and immune responses. The activation of estrogen receptors (ERα and ERβ) and other steroid hormones, including progesterone, is strongly connected to the signaling pathway linked to breast cancer and reproductive tumors [105]. Because estrogen stimulation triggers signaling cascades that increase the expression of genes that promote proliferation, survival, and invasiveness, this pathway is especially important in hormone-dependent malignancies such as estrogen receptor-positive (ER+) breast cancer. For example, cross-activation with the PI3K/AKT/mTOR pathway enhances proliferation and the epithelial-to-mesenchymal transition (EMT) [106]. In tumor cells, Hedgehog signaling preserves stem cell characteristics, allowing for metastasis and regeneration. The pathway, which is triggered by Hedgehog ligands interacting with PTCH1 and SMO, uses GLI factors to induce transcriptional activity [107]. In solid tumors like pancreatic cancer and medulloblastoma, hedgehog signaling is a prospective therapeutic target since it is often overactivated in malignancies and promotes tumor growth, drug resistance, and stromal interactions [108]. One important signaling mechanism that connects cancer cells and the immune system is the JAK/STAT pathway. It causes the phosphorylation of JAK proteins, which in turn activates STAT TFs, and is triggered by cytokines such as interleukins and interferons [109]. Chronic activation of the JAK/STAT pathway in cancers promotes immune evasion, creates a pro-inflammatory environment, and drives tumor progression, affecting immune cells and distant sites such as lymph nodes and bone marrow [110].

Epigenetic changes that modify gene expression without directly altering the DNA sequence are influenced by the chromatin remodeling and transcription regulation pathway. Chromatin remodeling complexes, including SWI/SNF, NuRD, and Polycomb, which control DNA accessibility for TFs and transcription complexes, are essential to this pathway [111]. When these complexes malfunction, oncogenes can be activated, and tumor suppressor genes can be repressed during carcinogenesis. For example, mutations in SWI/SNF complex subunits, including ARID1A or SMARCA4, are prevalent in a variety of tumor types and enhance the tumor’s epigenetic flexibility and adaptability. Additionally, changes in the enzymes responsible for histone modifications (HDACs, HATs, and PRMTs) and DNA methylation (DNMTs) can provide an epigenetic environment that promotes the growth of tumors [112]. By acting directly on chromatin and encouraging connections with TFs and RNA polymerase II, the transcriptional co-activation pathway, in contrast, coordinates the activity of coactivators and multiprotein complexes that control gene transcription. The transcription of important genes involved in angiogenesis, tumor survival, and cell proliferation can be amplified by coactivators like Mediator, TRRAP, and CBP/p300 [113]. Lastly, Krüppel-associated box domain-containing zinc-finger proteins (KRAB-ZFPs) have been shown in recent studies to play a function in preserving genomic stability and regulating stress responses in cancer cells [114]. An example comes from diffuse large B-cell lymphoma (DLBCL), a group of KRAB-ZFPs unique to primates are elevated. A worse prognosis and more genomic instability are linked to this overexpression. Two KRAB-ZFPs, specifically ZNF587 and ZNF417, were found in target evolutionarily recent transposable elements. When these proteins were depleted in DLBCL and other tumor cell lines, heterochromatin redistribution, replicative stress, and cGAS–STING pathway activation occurred, which in turn triggered an inflammatory response mediated by interferon. The cells were more vulnerable to macrophage-mediated phagocytosis because of this reaction, which also improved neoantigen presentation [115]. In DLBCL, KRAB-ZFPs maintain heterochromatin and suppress replicative stress-induced immune surveillance, promoting tumor growth and immune evasion. However, their precise timing and spatial behavior under osmotic stress in tumor cells remain unclear and require further study (Figure 2).

### 2.3. Key TFs as Regulatory Hubs in Oncogene Promoter Dysregulation

Through the investigation of TFBS, we identified a subset of transcription factors binding multiple promoter regions, indicating their role as central regulatory nodes within oncogene expression networks. This integrative analysis elevates motif discovery to a systems-level perspective, uncovering an interconnected hub of eleven TFs with MYC as a central node. Such emergent properties, not predictable from existing literature, underscore the originality of our computational approach. STRING network analysis further confirmed this cooperative regulatory module, revealing a densely connected cluster of eleven TFs sharing common binding sites. Many of these TFs are known to participate in critical pathways related to cancer development, including inflammation and cellular stress responses, indicating their possible contribution to oncogenic processes. The following sections will detail these TF hubs and explore their functional interactions and implications in oncogene regulation (Figure 3).

Among the identified TFs, TAF1 is an essential component of the basal TF complex TFIID, crucial for starting gene transcription by RNA polymerase II [116]. It acts as a scaffold and has enzymatic functions that regulate gene expression and the cell cycle [117]. Overexpression or dysregulation of TAF1, driven by genetic and epigenetic changes, was found in NSCLC, where it activates TGFβ1 [118] and is associated with progression of human prostate cancers to the lethal castration-resistant state, is common in various cancers [119]. Notably, external factors, like viral infections, can contribute to the development of some malignancies. As an example, HPV1’s E2 protein enhances TAF1 expression, increasing its activity [120]. An inhibitor of TAF-1, BAY 299 is able to induce cell death in acute myeloid leukemia cells, demonstrating its broad therapeutic potential across different cancer types [121]. BAY 299 can also improve the efficacy of bispecific anti-PD-1/PD-L1 antibodies by stimulating immune cells and encouraging functional transitions, especially in high-grade serous ovarian cancer. This immunomodulation happens through chromatin modification, which enhances anticancer responses [122].

The TF WT1 (Wilms’ Tumor 1) also exhibited a strong connection with TFs described in this report. Besides Wilms’ tumors, a kind of pediatric kidney cancer, other malignancies, including leukemia and some ovarian tumors, are linked to its altered expression. WT1′s tumor suppressor activity may be compromised by loss-of-function mutations, which would promote unchecked growth [123].

KLF6 (Krüppel-like factor 6) is a well-studied member of the SP/KLF TF family, characterized by a zinc finger structure, and it regulates diverse cellular biological processes including tissue growth and development, cell proliferation and differentiation, and vascular remodeling. Increasing evidence supports its role as a tumor suppressor, although the precise mechanisms of its anticancer effects remain not fully understood. Some classic pathways described for KLF6 tumor suppressor activity include activation of the cyclin-dependent kinase inhibitor p21 through a p53-independent mechanism, reduction of the cyclin D1/CDK4 complex via direct interaction with cyclin D1 leading to cell cycle arrest, inhibition of the proto-oncoprotein c-Jun activities, downregulation of VEGF expression, which impacts angiogenesis, upregulation of E-cadherin associated with maintaining cell adhesion and inhibiting metastasis, and induction of apoptosis in cancer cells [124]. In the same TF family, the analysis highlighted the presence of KLF15 (Krüppel-like factor 15). Its dysregulation is associated with malignancies such as lung adenocarcinoma, gastric cancer, breast cancer, and colorectal cancer by exerting anti-tumor effects in breast cancer by inhibiting tumor cell proliferation and migration, inducing cell cycle arrest, and promoting apoptosis [125].

ZNF281 is a zinc finger TF that plays a vital role in regulating cell differentiation and proliferation. It has been implicated as an important regulator of epithelial–mesenchymal transition (EMT), cancer stemness, and metastasis in various cancers, including colorectal, breast, and pancreatic carcinomas [126]. In hepatocellular carcinoma ZNF281 promotes tumor progression by suppressing mitochondrial biogenesis and function via repression of key mitochondrial TFs like NRF1 and PGC-1α [127]. This repression facilitates cancer cell invasion and metastasis. Furthermore, in primary CRC samples, expression of ZNF281 increased during tumor progression and correlated with recurrence [128].

EGR1 also came out from this analysis as a TF able to bind multiple TFBS, it is composed of activation and repression regulatory regions and zinc finger domains that regulate gene transcription, playing key roles in cell proliferation, differentiation, invasion, and apoptosis [129]. Its expression varies across tumor types, acting as an oncogene in prostate and gastric cancers, where higher EGR1 levels correlate with increased malignancy and poor prognosis. This finding is supported by clinical data; for instance, in gastric cancer, high EGR1 expression correlates significantly with advanced tumor stage (Spearman’s ρ ~ 0.6, *p* < 0.001) and poor prognosis [130,131,132]. Conversely, in gliomas and melanocytomas, EGR1 promotes tumor suppressor gene p21Waf1/Cip1 expression and induces apoptosis, functioning as a tumor suppressor in these cancers [133].

Patz1, also known as Zfp278 or MAZ-related factor (MAZR), is a POZ, AT-hook, and Kruppel zinc finger protein that modulates reprogramming efficiency depending on the cellular context [134]. PATZ1 overexpression is observed in several tumor types, including breast, colon, and prostate cancers, where it supports oncogenic transcription programs and enhances cell proliferation and survival [135,136,137]. Conversely, it was also demonstrated that mice lacking PATZ1 develop lymphomas and other tumors, suggesting that PATZ1 functions as a potential tumor suppressor in lymphomagenesis and possibly in other cancers [138].

Myc-associated zinc finger protein (MAZ) is a widely expressed TF and has a crucial role in regulating gene transcription related to cancer progression, including proliferation, apoptosis, and angiogenesis [139]. MAZ promotes tumor-associated angiogenesis by controlling genes like VEGF; its repression can be induced by factors such as oxidative stress and epigenetic changes, sustaining tumor growth and blood supply through pathways like STAT3 signaling [140]. Its overexpression has been recently associated with several cancers [141,142].

ZSCAN22 is a zinc finger and SCAN domain TF that helps maintain cellular homeostasis by regulating genes involved in proliferation, differentiation, and apoptosis [143]; genetic and epigenetic alterations disrupting its function can impair critical pathways like cell cycle and DNA repair, contributing to tumor progression [144], while non-coding RNAs further modulate its expression in complex regulatory networks [145].

GA-binding protein alpha (GABPA) is an ETS family TF that forms a functional heterotetramer with the GABPβ subunit to regulate gene expression essential for mitochondrial biogenesis, cellular respiration, and energy homeostasis [146,147]. In cancer, GABPA expression and activity are modulated by pathways such as PI3K/AKT/mTOR and AMPK that control tumor growth and metabolism; overexpression supports tumor proliferation and angiogenesis by enhancing energy production, while epigenetic silencing impairs mitochondrial gene activation, reducing tumor adaptability to hypoxia and shifting metabolism towards HIF-1α-driven glycolysis [148].

The intricate interplay of the 11 TFs provides a comprehensive picture on the molecular pathways driving cancer formation (Appendix A). The STRING database analysis demonstrates that nine of the eleven TFs form a highly interconnected network, with MYC serving as the central hub, emphasizing its critical role in transcriptional control and carcinogenic signaling pathways. Altogether, these findings emphasize that even minor perturbations in the expression or activity of these TFs can disrupt cellular homeostasis, potentially initiating or promoting tumorigenesis. Further experimental studies are required to explore the functional consequences of these interactions and the lack of connectivity between specific TFs (Figure 3).

## 3. Discussion

Multicellularity constitutes a significant evolutionary challenge, necessitating a balance between cellular proliferation, essential for growth, regeneration, and response to damage, and the stringent regulation of mutations to avert the conversion of normal cells into tumorigenic cells. Multicellular organisms have evolved exceptionally sophisticated mechanisms to preserve genetic stability and mitigate the risk of tumorigenesis, implementing strategies tailored to the specific requirements of each tissue type and adapting them to physiological demands and selective pressures. In tissues with high turnover, such as the intestinal epithelium and the hematopoietic system, the balance between proliferation and mutation control is maintained through a dynamic, hierarchical structure. In hematopoiesis, a clear cellular hierarchy enables stem cells to distribute the proliferative load among progenitor populations [149]. This method limits the accumulation of mutations, preserving genome integrity even during extensive cell divisions. In contrast, the colon epithelium undergoes rapid turnover, facilitating the early elimination of mutated cells through a process of “washing out”. This tactic reduces the time available for further genetic alterations to accumulate but requires a high rate of stem cell renewal, exposing the system to potential genomic instability if surveillance mechanisms fail. On a genomic level, cancer is influenced by a broad range of mutational processes. Most somatic mutations are classified as passenger mutations, which are functionally neutral and often arise from imperfect DNA repairs or exposure to carcinogens. However, a small subset of mutations, known as driver mutations, undergoes positive selection and confers adaptive advantages to tumor cells, promoting growth, survival, and invasive capabilities [150]. A significant recent concept in cancer biology is the identification of “goners.” These are genes or mutations that, unlike driver mutations, do not directly initiate or sustain malignant transformation. Instead, goners contribute to tumor progression by indirectly supporting cancer’s growth and survival. They achieve this by reshaping the tumor microenvironment, influencing metabolic pathways, or enhancing the tumor’s access to vital resources such as nutrients and oxygen. Understanding the role of goners adds a new dimension to cancer dynamics, highlighting how these auxiliary players enhance tumor adaptability [151]. Among the major factors driving tumor progression, TF dysregulation is particularly prominent. Our study contributes at three distinct levels: (i) new data—the identification of five recurrent binding motifs and 128 candidate TFs regulating 622 cancer driver genes; (ii) new analysis—the reconstruction of an interaction network that revealed an emergent MYC-centered hub of eleven TFs; and (iii) new concept—the Eternity Triangle, a framework explaining how TF dysfunction simultaneously drives uncontrolled proliferation, impaired differentiation, and genomic instability. Collectively, these findings demonstrate both the methodological rigor and conceptual innovation of our work. TFs, as master regulators of transcription, are frequently altered by somatic mutations, epigenetic modifications, or viral infections. Our in silico analysis identified eleven key TFs sharing common binding sites, suggesting their capacity to induce temporal and spatial alterations recurrent across tumor types. TAF1 stands out as a critical regulator within transcriptional networks. When overexpressed, as observed in cervical and hepatocellular carcinomas, it drives angiogenesis and cellular proliferation. In contrast, under expression of TAF1, often modulated by microRNAs, reduces the activity of tumor suppressor genes, creating a tumor-friendly environment. Given its capacity to indirectly alter tumor adaptability, TAF1 might be seen as an ideal example of a goner. Another key TF, PATZ1, exhibits a dual role: when overexpressed, it amplifies gene dysregulation, and when under expressed, it compromises transcriptional control, promoting uncontrolled growth. A similar mechanism is observed with MAZ, which, by activating VEGF and supporting angiogenesis, ensures that the tumor has a supply of nutrients and oxygen, providing an adaptive advantage, even though it is not a direct driver. These examples illustrate how these factors, while not directly responsible for malignant transformation, contribute to the consolidation and progression of tumors. Moreover, a key factor in the dysregulation of TFs is viral oncogenesis. Through epigenetic changes, oncoviruses like HPV, EBV, and HBV can change how TF functions. For example, HPV16′s E2 protein promotes carcinogenic transcription by upregulating TAF1 expression. The disruption of cellular homeostasis caused by aberrant promoter methylation is another way that EBV and HBV affect WT1 and other TFs. Viruses’ capacity to alter TF activity emphasizes how they influence the tumor microenvironment. These infections can produce an epigenetic environment that is conducive to unchecked cell division, angiogenesis, and immune evasion by focusing on regulatory pathways.

The exploration of transcriptional regulatory networks mediated by TFs, together with an improved understanding of the interplay between driver and passenger mutations, holds great promise for the development of innovative therapeutic strategies. Targeting hyperactive TFs with selective inhibitors, or reactivating dysfunctional TFs, represents a compelling avenue for mitigating tumor progression. Although this study provides a pan-cancer map of putative TF regulators, the prognostic and mechanistic relevance of central hubs, particularly MYC, TAF1, and EGR1, requires validation in future studies using independent, cancer type–specific cohorts such as those available from TCGA. In addition, therapeutic approaches aimed at disrupting key regulatory interactions, including those involving TAF1, MYC, MAZ, and other TFs such as KLF15 and ZNF281, may suppress tumor growth and modulate stress adaptation, thereby enabling more precise interventions against cancer. Such strategies could minimize off-target effects while effectively targeting transcriptional networks that sustain tumorigenesis. Beyond their value as direct therapeutic targets, TFs can also be identified and characterized as biomarkers to prevent tumor progression, enhance responses to existing treatments, and inform personalized therapies tailored to the functional biology of the tumor as well as its genetic alterations. Collectively, these discoveries may represent a major advance in cancer research, transforming the complexity of molecular regulatory networks into a powerful tool against the disease itself. Nevertheless, tissue-specific propensities to mutation remain among the most enigmatic challenges in cancer biology.

## 4. Materials and Methods

Four bioinformatics tools were included in the analytic pipeline to find common TFs linked to the gene promoters listed in the IntOGen database Release v2024.09.20 (IntOGen, BBGLab, Barcelona, Spain). With its current information on genes involved in carcinogenesis, this database is a good place to start when looking for promoters that might be connected to processes relevant to cancer [152,153].

### 4.1. Data Collection and Gene Promoter Analysis

We retrieved the promoter sequences for all 622 target genes in FASTA format. These genes were not arbitrarily selected; they represent a rigorously curated, pan-cancer set of driver genes provided by the IntOGen database. Using this well-defined set of effectors, we adopted a reverse-engineering strategy to move upstream and identify the common transcriptional regulators orchestrating their expression. This unbiased approach enhances reproducibility and ensures that the analysis focuses on functionally validated drivers rather than context-dependent passengers. Promoters were defined as the region extending from 1000 base pairs upstream to 100 base pairs downstream of the transcription start site (TSS). The corresponding Ensembl Gene IDs were obtained from IntOGen [154], and nucleotide sequences were extracted from the Ensembl genome browser (GRCh38.p14 build), a gold standard in genomic annotation. This resource integrates extensive computational and manual curation, guaranteeing sequences free of common artifacts and accurately aligned with the reference genome. Consequently, no additional preprocessing or quality control steps, such as adapter trimming or Q30 filtering, were required, since the analysis relied on these pre-curated canonical sequences [155]. IntOGen was selected as the primary source of cancer-associated genes instead of relying exclusively on TCGA (The Cancer Genome Atlas). While TCGA offers extensive raw multi-omics datasets and patient-level annotations, its direct use requires substantial preprocessing and the construction of custom pipelines to distinguish driver from passenger mutations. IntOGen, by contrast, integrates data from TCGA, ICGC, and other large-scale sequencing consortia, applying robust statistical algorithms such as OncodriveCLUST and OncodriveFM to systematically prioritize cancer drivers. This curated, pan-cancer perspective reduces noise, highlights recurrent alterations of greater biological and clinical significance, and provides a harmonized framework for reconstructing transcriptional regulatory networks and identifying core oncogenic pathways. Accordingly, IntOGen offered a more efficient and biologically meaningful foundation for in silico analyses.

Promoter sequences were then analyzed using MEME-ChIP, a component of the MEME Suite specifically designed for motif discovery in DNA datasets originally developed for ChIP-seq and CLIP-seq applications. All tools were executed via their web-accessible servers under default configurations to ensure maximum reproducibility and accessibility. MEME-ChIP (v5.5.0) was employed to detect both novel and known motifs in the selected promoters, enabling the identification of regulatory elements recognized by TFs. For motif discovery, the algorithm was configured to search for up to five motifs, with widths ranging from six to twenty nucleotides. The decision to expand beyond the default of three motifs was methodological: too few motifs increase the risk of selection bias by favoring highly conserved patterns, potentially missing less dominant but biologically important motifs, whereas too many increase the likelihood of spurious results, lowering the signal-to-noise ratio. Preliminary testing indicated that six motifs produced redundant outputs, whereas five represented an optimal compromise, balancing sensitivity and specificity, maintaining control of the false discovery rate, and ensuring robust detection of statistically significant motifs across heterogeneous promoter regions [156,157].

Motifs identified by MEME-ChIP were subsequently characterized with Tomtom (v5.5.0), which compares query motifs against libraries of known binding profiles. The HOCOMOCO v11 human database, derived largely from ChIP-seq experiments, served as the reference. The Benjamini–Hochberg method is used to determine the q-value, which is the lowest false discovery rate that permits a similarity to be deemed significant [35]. For this analysis, a q-value threshold of <0.05 was applied to define statistically significant matches. Each discovered motif was compared against the database, including reverse complement matches where applicable, with outputs consisting of ranked matches by *p*-value, quantitative alignments, and visual motif logos. Q-values indicated the minimum false discovery rate required for significance, thereby ensuring high confidence in motif-TF associations. The final product was a curated list of TFs, each supported by conserved DNA-binding domains, forming the basis for downstream regulatory network reconstruction [158] (Figure 4).

### 4.2. Data Comparison and Reporting

The TFs identified from motif analysis were further examined using STRING (Search Tool for the Retrieval of Interacting Genes/Proteins, v11.5), a platform designed to assess both functional and physical protein–protein interactions. STRING integrates diverse evidence sources, including literature mining, curated databases, sequence homology, evolutionary conservation, and experimental validation. Each predicted interaction is assigned a confidence score representing its likelihood of biological significance. In this study, the network was generated using a high-confidence threshold of >0.700, consistent with STRING’s definition of “high confidence.” This filtering ensured that only robustly supported interactions were retained, thereby increasing the reliability of downstream analyses aimed at reconstructing transcriptional and signaling networks relevant to cancer biology [159]. 

**Figure 4 ijms-26-09933-f004:**
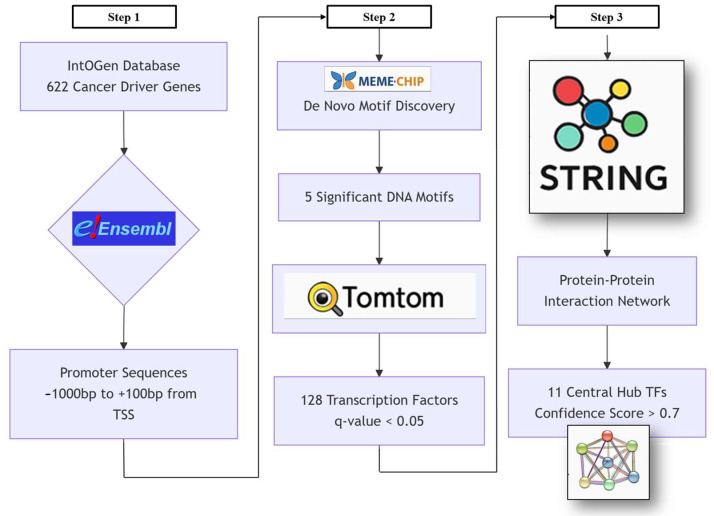
Comprehensive bioinformatics workflow for identifying TFs regulating 622 cancer driver gene promoters. Step 1: A curated list of 622 driver genes was obtained from IntOGen, which integrates pan-cancer data. Promoter regions were defined as −1000 bp to +100 bp relative to the transcription start site (TSS), and canonical sequences were retrieved from Ensembl (GRCh38.p14) in FASTA format. Step 2: Motif discovery was performed with MEME-ChIP (v5.5.0), configured to identify up to five motifs with widths of 6–20 nucleotides, balancing sensitivity and specificity. Discovered motifs were compared against the HOCOMOCO v11 database using Tomtom (v5.5.0), with significance thresholds of E-value < 0.05 and q-value < 0.1. This yielded statistically significant matches linking the motifs to 128 TFs characterized by conserved DNA-binding domains. Step 3: The identified TFs were analyzed in STRING (v11.5) to reconstruct protein–protein interaction networks, applying a high-confidence interaction threshold (>0.700).

## 5. Conclusions

The complex molecular harmony coordinated by TFs and their dysfunction disrupt the cellular life cycle, altering DNA replication, cell proliferation, and the onset of somatic mutations toward a fate of transformation. Understanding these dynamics allows us to restore genomic order in a universe of molecular chaos, stemming the inevitable flow toward cancer evolution. The findings provide a map that unveils the fundamental molecular principles leading to tissue vulnerability and the emergence of mutations. In the future, it will be essential to approach this challenge as a multidimensional mosaic, mapping TF binding sites precisely and intertwining genomic, transcriptomic, and proteomic data. Only through this synthesis can we gain a deeper and more unified vision of the role of TFs in the complex narrative of cancer, opening new avenues for interventions that restore tissues to their original balance.

## Figures and Tables

**Figure 1 ijms-26-09933-f001:**
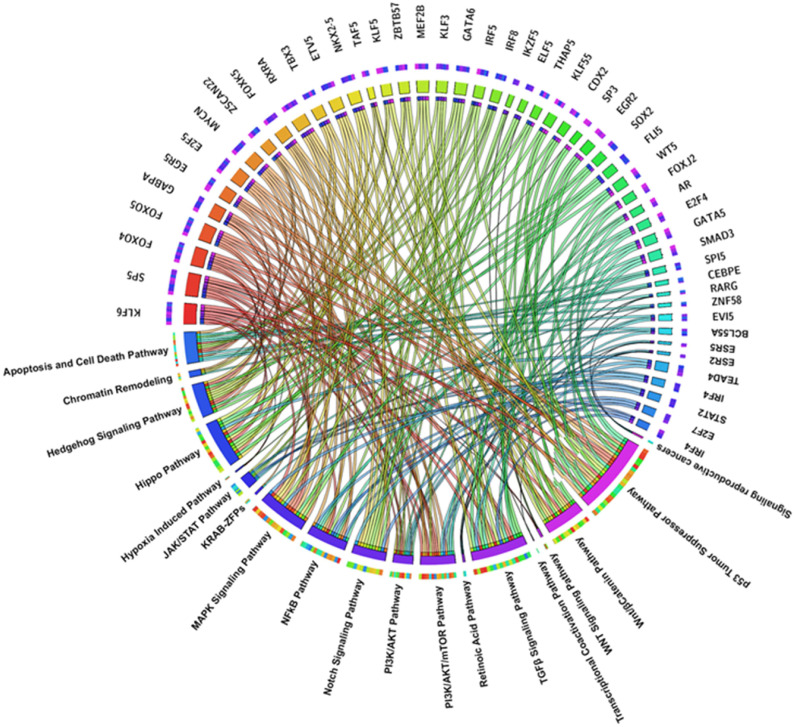
The circos plot depicts the relationship between the different signaling pathways and TFs. The area of each colored ribbon depicts the proportion of the signaling pathway contributes to a particular clinicopathological category.

**Figure 2 ijms-26-09933-f002:**
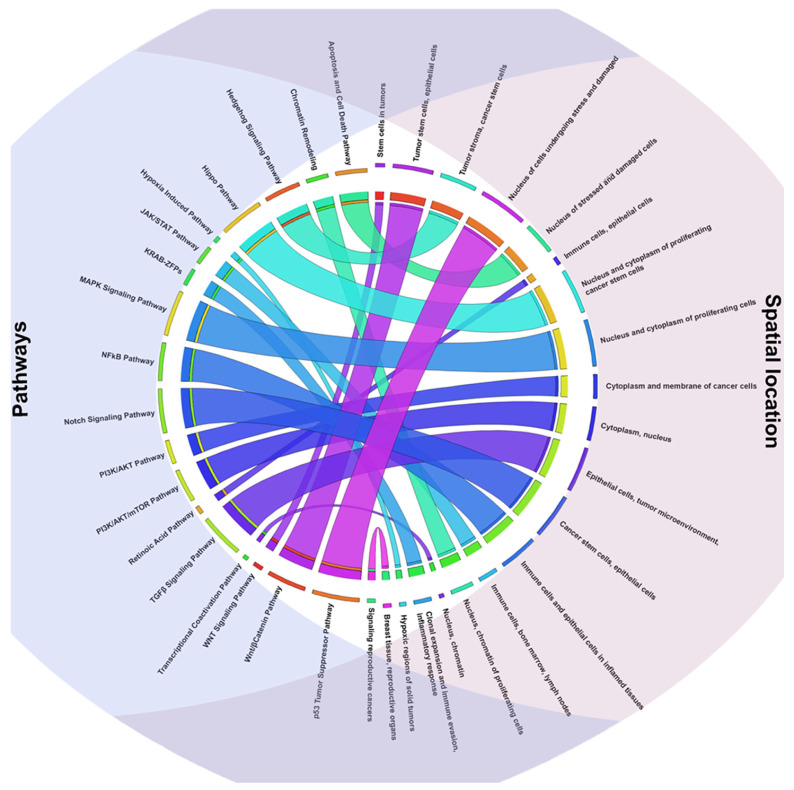
The circos plot depicts the relationship between the different signaling pathways and the spatial location.

**Figure 3 ijms-26-09933-f003:**
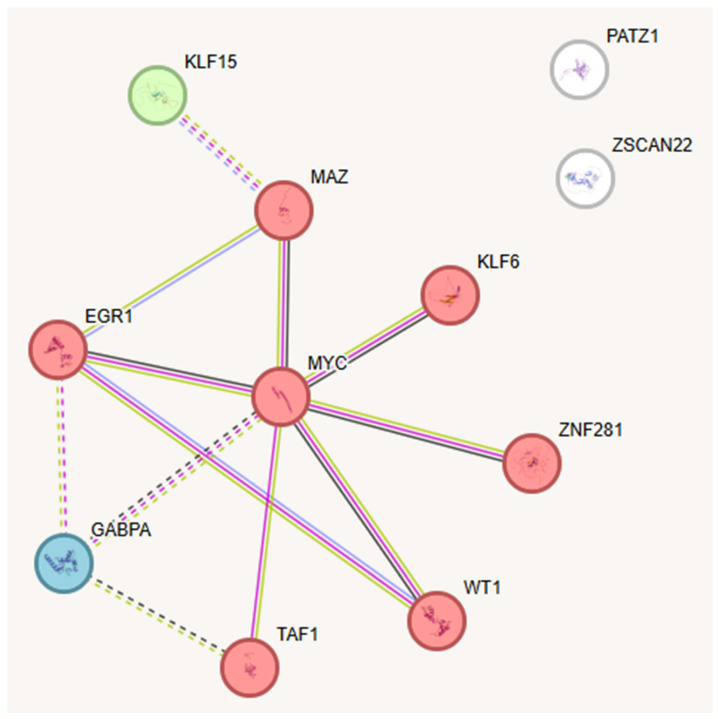
STRING-derived protein–protein interaction network of TFs binding multiple promoter regions in cancer-associated genes, revealing potential central regulatory hubs. The network comprises 11 TFs, with MYC emerging as the main hub, extensively connected to EGR1, MAZ, KLF6, ZNF281, WT1, TAF1, and GABPA, forming a tightly interlinked cluster indicative of cooperative regulation. KLF15, PATZ1, and ZSCAN22 appear more peripheral or disconnected.

**Table 1 ijms-26-09933-t001:** De novo motifs identified in promoters of 622 cancer driver genes and their TFs. The five DNA motifs found by MEME-ChIP analysis are shown in the table in logo format, together with their matching E-values and the human TFs that they significantly resemble, as determined by Tomtom comparison with the HOCOMOCO v11 database. The E-value represents the statistical significance of each motif’s enrichment, with lower values indicating a higher confidence that the motif is not present by chance.

MEME-ChIP Motif (Sequence Logos)	E-Value	Human TFs
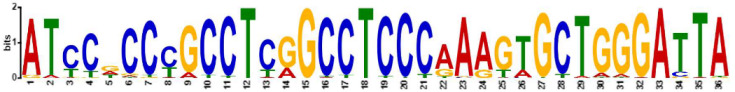	**3.4 × 10^−143^**	ZNF770, IKZF1, WT1, ZNF263, KLF6, ZSC22, ZN281, ZNF250, TEAD4, GFI1, EGR1, PATZ1, ELF5, ZNF341, GFI1B, ZNF467, MAZ, KLF15, ZNF322, TAF1, GABPA, SP3, SP1, TFAP4, DUX4
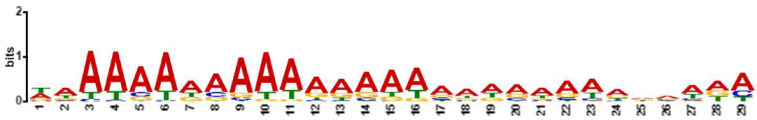	**6.3 × 10^−140^**	AIRE, ALX1, ANDR, BC11A, CDX1, CDX2, ELF3, EVI1, FOXJ2, FOXJ3, FOXK1, FOXO1, FOXO4, FOXQ1, GATA3, GATA6, HNF6, IRF1, IRF2, IRF3, IRF4, IRF8, LEF1, LHX3, MEF2A, MEF2B, MEF2D, NFAC1, NKX61, NR2E3, PIT1, PO2F1, PRDM6, SOX2, SOX4, SOX5, SPI1, SPIB, SRF, SRY, STAT2, TCF7, Z354A, ZFP28, ZFP82, ZIM3, ZN394, ZNF260, ZNF354A, ZNF394, ZNF85,
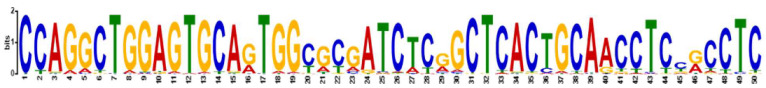	**6.6 × 10^−168^**	THAP1, TEAD1, RARG, ZSC31, ESR2, ESR1, CEBPE, ZNF257, ZNF18, MYB, TBX3, ETV5, NKX21, TAF1, SMCA5, NKX25
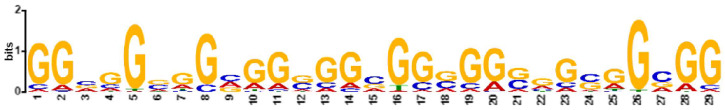	**3.8 × 10^−153^**	PATZ1, SP2, SP1, SP3, ZN467, WT1, MAZ, VEZF1, KLF15, ZN341, EGR1, KLF6, ZN263, KLF3, SP4, EGR2, ZBT17, ZN770, ZN281, PTF1A, TAF1, E2F6, KLF9, RXRA, THAP1, E2F1, ASCL1, KLF1, FLI1, E2F4, MYOD1, KLF12, E2F7, COT1, MBD2, MXI1, MYOG, SRBP2, USF2, TFDP1, GABPA, NR1H4, SALL4, RFX1, ZSC22, ZN335, CTCFL, MYCN
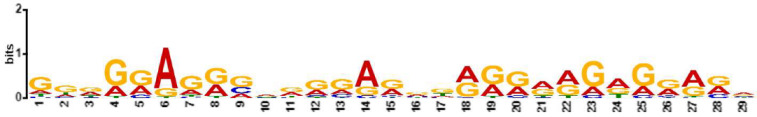	**4.6 × 10^−141^**	MAZ, ZN467, VEZF1, ZN263, FLI1, WT1, ZN341, PATZ1, KLF15, IRF3, ZBT17, RXRA, BC11A, ETS2, E2F6, E2F1, ETV5, SPI1, SPIB, ELF5, SP3, EGR2, PTF1A, SP4, TFDP1, ZN281, GATA2, E2F4, E2F7, SP2, OLIG2, KLF6, TBX3, SP1, ETV2, SALL4, KLF3, IRF4, ERG, E2F3, SOX2, EGR1, SMAD3, GATA1, ZN770, NR1D1, TAL1, ETV4, ETS1, IRF8, PRDM6, ZFP82, COT1, RARA, SOX4, ELF3, PRDM1, NFAC1, FOXO1, MZF1, ZSC22, GABPA, ZNF41, IRF2, ZN418, TBX21, TAF1, SMAD2, ZN816, NKX25, IRF1, ZN586, ZN768, ELF2, NFIC, SRBP2

**Table 2 ijms-26-09933-t002:** The table summarizes major transcription factor (TF) families identified, listing key members, their main biological functions, and roles in cancer as oncogenes or tumor suppressors. These TFs regulate critical processes such as cell cycle, apoptosis, proliferation, differentiation, angiogenesis, immune response, EMT, DNA repair, and cancer stem cell maintenance. Dysregulation of these TFs through mutation or expression changes contributes to cancer development, progression, metastasis, and therapy resistance in various cancers like breast, lung, prostate, liver, colon, pancreatic, hematologic, and sarcoma. Some TF families have context-dependent roles, acting as oncogenes or tumor suppressors depending on the tumor type and molecular environment.

TF Family	Representative Members	Main Functions	Oncogenic/Tumor Suppressor Role	Associated Cancers/Implications
E2F Family	E2F1, E2F3, E2F4, E2F6, E2F7	Regulate genes essential for G1→S phase transition in the cell cycle	Frequently oncogenic when deregulated; loss of regulation leads to uncontrolled proliferation	Retinoblastoma, breast cancer, multiple carcinomas
MYC Family	MYCN	Controls proliferation, metabolism, and interaction with PI3K/AKT and CDKs	Oncogenic; amplification linked to poor prognosis	Neuroblastoma, lung carcinoma, lymphoma, breast cancer
KLF Family	KLF1, KLF3, KLF6, KLF12, KLF15	Regulate apoptosis, oxidative stress, angiogenesis, and proliferation	KLF6 as tumor suppressor; KLF3 and KLF15 can be oncogenic	Prostate, liver, colon cancers
FOXO Family	FOXO1, FOXO3, FOXO4	Control apoptosis, DNA repair, and stress response	Tumor suppressors; often inactivated via AKT phosphorylation	Skin, liver, lung, prostate cancers
SP Family	SP1, SP3	Regulate genes for proliferation, DNA synthesis, stress response	Oncogenic when overexpressed	Breast cancer, chemoresistance
GATA Family	GATA2, GATA3, GATA6	Control EMT, migration, metastasis, immune regulation	Context-dependent; can promote or suppress tumors	Breast cancer, lung adenocarcinoma, AML
STAT Family	STAT3	Regulate proliferation, survival, immune evasion	Oncogenic; promotes NK cell evasion	Breast, lung, melanoma
IRF Family	IRF1, IRF8	Regulate apoptosis, immune activation	Tumor suppressors in immunity	Various cancers; loss promotes immune suppression
NF-κB Family	NFAC1, NF-κB1	Promote inflammation, proliferation, metastasis	Oncogenic when persistently active	Skin, colon, breast cancers
SOX Family	SOX2	Maintain stemness, plasticity, therapy resistance	Oncogenic in stem cell maintenance	Lung carcinoma, glioblastoma
HOX Family	HOXA9	Regulate differentiation, proliferation	Oncogenic in leukemia and solid tumors	Leukemia, various solid tumors
SMAD Family	SMAD4	TGF-β signaling, apoptosis, differentiation	Tumor suppressor; loss promotes immune evasion	Pancreatic cancer, others
Sex Hormone Receptors	AR, ERα, ERβ	Regulate hormone-dependent growth	Oncogenic in deregulated states	Prostate, breast, ovarian, endometrial cancers
ETS Family	ETS1, ETS2	Regulate proliferation, migration, angiogenesis	Oncogenic	Multiple cancers, immune modulation
ZNF Family	ZNF18, ZNF250, ZNF560	Gene regulation, tumor behavior modulation	Oncogenic when overexpressed	Osteosarcoma
PRDM Family	PRDM1, PRDM6	Epigenetic regulation, gene repression	Oncogenic; immune evasion	HCC, medulloblastoma
Retinoic Receptor Family	RARA, RARG	Regulate differentiation, proliferation, apoptosis	Disrupted function oncogenic	APL, skin, lung cancers
EGR Family	EGR1, EGR2	Control proliferation, apoptosis, immune response	Dual role; tumor suppressor or oncogenic depending on context	Multiple cancers, therapy resistance
MEF2 Family	MEF2A, MEF2B, MEF2D	Regulate differentiation, metabolism, angiogenesis	Oncogenic when dysregulated	Sarcomas, solid tumors

## Data Availability

Data is contained within the article and Appendix A.

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
