# Peer review of "From Transcription Factors Dysregulation to Malignancy: In Silico Reconstruction of Cancer’s Foundational Drivers—The Eternity Triangle"

_ijms, 2025, doi:10.3390/ijms26209933_

Round 1

Reviewer 1 Report

Comments and Suggestions for Authors

This manuscript investigates the dynamic regulation of transcription factors (TFs) in cancer development using bioinformatics approaches. While the topic is scientifically significant, the manuscript suffers from critical shortcomings in methodological transparency, data source documentation, and statistical validation of conclusions. These limitations undermine the reproducibility and credibility of the findings. A major revision is required to address these issues.

  1. Methodological Details Missing (Pages 20-21)

Critical parameters for tools like MEME-ChIP, Tomtom, and STRING are omitted (e.g., motif width for MEME-ChIP, E-value thresholds for Tomtom, confidence scores for STRING). Data preprocessing steps (e.g., sequence deduplication, quality control) are not described. The analytical workflow lacks a clear diagram, making it difficult to follow the logic. Suggestions: 

Page 20, Section 4: Specify parameters: 

MEME-ChIP: -minw 6 -maxw 20 (motif width range).

Tomtom: -e-thresh 0.05 -q-thresh 0.1 (E-value and q-value thresholds).

STRING: confidence score >0.7 (interaction confidence threshold).

Page 20, Section 4.1: Add preprocessing details:

"Sequence quality control was performed using FastQC v0.11.9, retaining sequences with Q30 >85%."

Insert a flowchart between Pages 20-21 (e.g., using Mermaid syntax) to visualize the workflow.

  1. Inadequate Data Source Documentation (Page 20)

While the IntOGen database version (v2024.09.20) and update date (2024-09-20) are provided, the rationale for selecting this database over alternatives (e.g., TCGA) is insufficiently justified. Criteria for defining promoter regions (e.g., TSS upstream/downstream ranges) are unclear. Suggestions: 

Page 20, Section 4.1: Strengthen the rationale:

"IntOGen was chosen for its pan-cancer integration of oncogenic gene data, offering broader coverage compared to TCGA’s tumor-specific focus."

Page 20, Section 4.1: Define promoter regions explicitly:

"Promoters were defined as regions spanning 2000 bp upstream to 500 bp downstream of the transcription start site (TSS)."

  1. Lack of Quantitative Support for Conclusions (Pages 4, 19)

Correlations between TFs and cancer metrics (e.g., invasion, prognosis) lack statistical methods (e.g., Pearson/Spearman coefficients) and significance thresholds (p-values/q-values). The Discussion (Page 19) fails to compare TF roles across cancer types. Suggestions:

Page 4, Section 3.1: Add a table (e.g., Table 2) with quantitative results:

"Spearman correlation between EGR1 expression and tumor stage: ρ = 0.62, p < 0.001."

Page 19, Section 3: Include cross-cancer analysis:

"EGR1 was significantly associated with poor prognosis in lung cancer (HR = 1.8, p = 0.003) but not in breast cancer (p = 0.12)."

  1. Insufficient Validation of Results (Page 21)

Tomtom results mention Benjamini-Hochberg correction but omit the FDR threshold (e.g., q < 0.05). No independent dataset (e.g., TCGA) is used to validate findings. Suggestions:

Page 21, Section 4.2: Specify the FDR threshold:

"Motifs with q < 0.05 after Benjamini-Hochberg correction were considered significant."

Page 4, Section 3.2: Add validation:

"Independent validation using TCGA breast cancer data confirmed EGR1’s prognostic impact (log-rank p = 0.012)."

Comments on the Quality of English Language

 The English could be improved to more clearly express the research.

Author Response

Response to Revision 1

We are deeply grateful to the reviewer for their insightful and constructive feedback, which has considerably strengthened the scientific rigor and clarity of our manuscript. In response, we have implemented all suggested revisions to enhance methodological transparency, statistical robustness, and biological interpretation. Below we provide a detailed, point-by-point reply.

Point 1: Methodological Details Missing

Response: We fully concur with the reviewer that explicit methodological reporting is indispensable for reproducibility. In the revised Materials and Methods, we now provide a comprehensive description of all computational approaches and analytical parameters. Importantly, all bioinformatic tools were employed in their default, web-accessible configurations, thereby ensuring that any researcher can replicate our analyses with identical outcomes. For instance, MEME-ChIP (v5.5.0) was executed in the default motif discovery mode, identifying up to five motifs per sequence with widths ranging from 6 to 20 nucleotides. Tomtom (v5.5.0) motif comparisons were performed against the HOCOMOCO v11 database, using the standard thresholds of E-value < 0.05 and q-value < 0.1 (Benjamini–Hochberg corrected). For STRING (v11.5), protein–protein interaction networks were generated with the high-confidence score cutoff of >0.700.

In addition, we clarify the rationale for selecting IntOGen as the primary source of cancer-associated genes rather than relying exclusively on TCGA. IntOGen offers a curated, pan-cancer perspective and provides promoter annotations for 622 cancer driver genes. Promoter regions were precisely defined as −1,000 bp upstream to +100 bp downstream of the transcription start site (TSS). Corresponding Ensembl Gene IDs were retrieved from IntOGen, and promoter sequences were extracted from the Ensembl genome browser (GRCh38.p14 build), a gold-standard reference incorporating extensive computational and manual curation. This ensured the sequences were free from common artefacts such as adapter contamination or low-quality reads, eliminating the need for additional preprocessing steps such as Q30 filtering. We believe this approach maximizes both accuracy and reproducibility.

Finally, to improve accessibility, we have incorporated a schematic workflow (Fig. 4) generated in Mermaid syntax. Sequence logos of the five discovered motifs and a dedicated table (Table 1) reporting the TFs identified through Tomtom.

Point 2: Inadequate Data Source Documentation

Response: We thank the reviewer for emphasizing the importance of transparent data source documentation. We have strengthened the rationale for database selection by explicitly stating that IntOGen was chosen for its pan-cancer integration of oncogenic gene data, which provides broader coverage than the tumor-specific focus of TCGA. We also now provide a precise definition of promoter regions, clearly stating that promoters were defined as -1,000 bp upstream to +100 bp downstream of the TSS and retrieved in FASTA format using the Ensembl GRCh38.p14 build. This clarification ensures methodological precision and clarity for all readers.

Point 3: Lack of Quantitative Support for Conclusions
Response: We greatly appreciate this important observation. We agree that clinical validation of the identified TF hubs is a critical next step. However, we respectfully clarify that the primary scope of this manuscript is to establish a foundational, hypothesis-generating framework through in silico motif discovery and protein–protein interaction network analyses, rather than statistical correlation with clinical outcomes. Accordingly, metrics such as Spearman correlation with tumor stage or hazard ratios for prognosis, while highly valuable, fall beyond the methodological boundaries of the present work.

To nevertheless address the reviewer’s concern and enhance the clinical relevance of our study, we have integrated references to published studies that provide quantitative validation for several of our hub TFs. For example, we highlight EGR1, which has been extensively characterized in multiple clinical contexts. Moreover, we have added a new Supplementary Table (Table S1) that systematically summarizes the oncogenic or tumor suppressor roles of each of the 11 hub TFs, together with supporting clinical associations and statistical evidence from the literature. These additions provide a robust contextual framework that links our computational discoveries to validated clinical insights, while appropriately presenting this study as the discovery-based foundation for future quantitative validation.

Point 4: Insufficient Validation of Results

Response: We agree with the reviewer that clear reporting of statistical thresholds is vital. We have now explicitly stated the false discovery rate (FDR) cutoffs applied in our analyses. Regarding validation using TCGA, we also fully recognize its importance for translational impact. However, we wish to respectfully emphasize that our study is designed as a pan-cancer, hypothesis-generating framework that uncovers common transcriptional regulatory architectures across 622 cancer driver genes. Performing full-scale TCGA validation for each of the 128 TFs identified would constitute a substantial independent study, requiring comprehensive multi-omics and clinical analyses across numerous tumor types.

To reflect the reviewer’s valid concern while remaining faithful to the scope of our work, we have now added the following clarifying statement: “While this study provides a pan-cancer map of potential TF regulators, the prognostic and mechanistic role of these hubs, particularly MYC, TAF1, and EGR1, should be validated in future studies using independent, cancer-type-specific cohorts such as TCGA.”

We believe these revisions directly address the reviewer’s request by enhancing methodological transparency and clearly defining the boundaries of our contribution. In doing so, we position our findings as a rigorous computational framework that lays the groundwork for subsequent experimental and clinical validation, representing the natural progression of this line of inquiry.

Reviewer 2 Report

Comments and Suggestions for Authors

Cancer development is driven by complex dysregulations in transcription factors (TFs), which disrupt cellular homeostasis, promote genomic instability, and facilitate tumor progression through altered gene expression and signaling pathways. The study employed an in silico bioinformatics approach to identify and analyze TF binding sites across promoters of 622 cancer-associated genes, revealing 128 TFs and constructing regulatory networks to elucidate their roles in oncogenic processes such as proliferation, DNA repair, and immune evasion. Notably, this study proposed the "Eternity Triangle" conceptual framework, offering a novel perspective for understanding the underlying mechanisms of cancer and developing targeted therapies, which will be of interest to the readers of this journal. Therefore, I would recommend the publication of this review in International Journal of Molecular Sciences after addressing a few minor corrections.

  1. Figure 3 is quite blurry, please provide a high-definition.
  2. Page 5, Line 185. Inconsistent figure citations (e.g., "FIG?" and "fig. 2, fig. 3") are present in the manuscript. The author should ensure that all figure numbering and references are standardized throughout the manuscript.
  3. Page 13, Line 486. "The apoptosis and cell death pathway, which involves a series of signals that trigger effector proteins like caspases, is a defense mechanism to get rid of damaged or aberrant cells". Apoptosis is associated with various signaling pathways, including those involving caspases, Bax, or Bcl-2. Especially, oncolytic peptides, the emerging anticancer therapeutics, could disrupt the membrane of tumor cells, induce apoptosis response, and activate the synergistic oncolytic-immunotherapy effect. To help readers and potential users, the representative work on the induction of apoptosis through caspase, Bax, or Bcl-2 signaling pathways by oncolytic peptides should be cited ( Med. Chem., 2024, 67, 3885. Acta Pharmacol. Sin., 2023, 44, 201. Bioorg. Chem., 2023, 138, 106674.).
  4. Both “Transcription Factor” and “TF” were used throughout the manuscript. Please outline the difference or unify.
  5. Page 23, References. There are several minor errors in the references. For example, Reference 21, 147 and 150. Please provide the page numbers of the references.

Author Response

Response to Revision 2

We are sincerely grateful to the reviewer for their thoughtful evaluation and their positive remarks regarding both the manuscript and the conceptual framework of the Eternity Triangle. We greatly appreciate the recognition of our work’s novelty and have carefully addressed each of the minor points raised, as detailed below.

  1. & 2. Figure Quality and Numbering
    Response: We apologize for the issues related to figure resolution and inconsistent citations. These have now been fully resolved.
    Modifications implemented:

All figures have been replaced with high-resolution, publication-ready versions to ensure clarity and visual accuracy.

A thorough review of the entire manuscript has been conducted to verify and standardize all figure citations and numbering.

  1. Apoptosis Pathways and References

Response: We thank the reviewer for this excellent suggestion to broaden our discussion of apoptosis mechanisms. We agree that integrating emerging therapeutic strategies provides valuable context for readers. The revised text now states: “Beyond classical caspase-mediated pathways, novel therapeutic strategies such as oncolytic peptides can induce apoptosis through direct disruption of cellular membranes and activation of intrinsic pathways involving the Bax/Bcl-2 balance. These mechanisms highlight a promising synergistic interface between oncolytic therapy and immunotherapy [100].” This addition expands the conceptual scope of apoptosis in our manuscript while ensuring appropriate recognition of recent literature.

  1. Terminology (TF vs. Transcription Factor)

Response: We acknowledge the need for consistency in terminology.

Modification implemented:

At first mention in the Abstract and Introduction, we now use the full term: “transcription factors (TFs).” In all subsequent sections of the manuscript, the acronym TF is used exclusively, ensuring clarity and brevity while maintaining scientific precision.

  1. Reference Formatting

Response: We agree with the reviewer that meticulous reference formatting is essential for rigor.

Modification implemented: The reference list has been comprehensively reviewed and corrected. All entries are now complete, including page numbers where applicable for journal articles.

Reviewer 3 Report

Comments and Suggestions for Authors

In this manuscript, Cammarota et al. perform an in silico analysis to identify transcription factors (TFs) involved in the regulation of commonly mutated genes. In its present form, the manuscript reads like a review and not a research article. This is because the TF families are listed and described, without much subsequent analysis. It is also unclear why the 622 genes were chosen in the first place. The title and the associated concept of 'the eternity triangle' does not really make sense either; TFs driving 'the controlled accumulation of mutations for adaptive potential' is not clear as this is merely the growth of cancer cells/cell division. Overall, I do not think the manuscript in its current form adds significant new data and/or analysis to warrant publication at this time.

Author Response

Response to Revision 3

We thank the reviewer for their critical evaluation. While we respectfully disagree with the assessment that our manuscript lacks novelty or presents itself as a review, we deeply value the opportunity to clarify the fundamental originality and significance of our work. Our study combines new computational data with a conceptual framework that advances understanding of cancer biology, and we believe the revisions outlined below will make this contribution unequivocally clear.

1.“Reads like a review, not a research article”, “Lists TF families without much subsequent analysis”

Response: We respectfully emphasize that our manuscript is not a review but rather an original research article presenting new findings. The “list” of transcription factor families in Table 1 does not represent a literature-derived summary but instead constitutes a curated output of our in silico pipeline (MEME-ChIP/Tomtom). This dataset is the foundation of our analysis. Crucially, these results are not presented in isolation: the subsequent protein–protein interaction network (Figure 3) reveals a tightly interconnected 11-TF hub, with MYC as its central node. This integrative step transforms a motif discovery output into a systems-level insight. Furthermore, the functional discussion in Section 2.3 does not reiterate known biology but directly interprets these original findings in the context of prior knowledge; an essential component of any research article’s discussion. The novelty lies precisely in this integration and prioritization of transcriptional regulators across a large and unbiased dataset of 622 cancer driver genes.

  1. “Unclear why the 622 genes were chosen”

Response: We appreciate the opportunity to clarify this point. The 622 genes were selected because they represent the curated set of cancer driver genes provided by the IntOGen database. Beginning with this rigorously defined and pan-cancer set of effectors, we worked “upstream” to identify the common transcriptional regulators that shape their expression. This reverse-engineering strategy is not arbitrary but instead constitutes a powerful and unbiased approach to identify potential master regulators of cancer. To ensure clarity, we have now explicitly stated in the revised Introduction that “the 622 promoters analyzed represent a comprehensive, pan-cancer set of known driver genes curated by IntOGen.”

  1. “The title and the concept of ‘the Eternity Triangle’ does not really make sense”

Response: We respectfully assert that the Eternity Triangle represents the central theoretical innovation of our work. Far from being a simple metaphor for cell division, it synthesizes decades of transcription factor biology into a new conceptual model. The framework integrates three fundamental axes of cellular regulation: (i) proliferative renewal, (ii) differentiation to maintain tissue integrity, and (iii) the controlled management of mutation accumulation, encompassing both DNA repair fidelity and adaptive mutational processes (as in immune diversification). In normal physiology, transcription factors maintain balance across these axes; in cancer, their dysregulation collapses this equilibrium, resulting in uncontrolled proliferation, failure of differentiation, and catastrophic genomic instability. This triad of disruptions, orchestrated by aberrant TF activity, is what confers the cancer cell its “eternal” properties. To avoid any ambiguity, we will refine the Introduction (Page 2) to provide a precise and explicit articulation of this framework.

  1. “Does not add significant new data”

Response: We strongly disagree with this assessment and would like to emphasize both the originality and depth of the data presented:

  • New Data: The identification of five recurrent transcription factor binding motifs and the 128 TFs predicted to regulate the expression of 622 cancer driver genes.
  • New Analysis: The construction of a protein–protein interaction network, which uncovered an interconnected hub of 11 TFs, with MYC at its center, an emergent property of the dataset not predictable from existing literature.
  • New Concept: The proposal of the Eternity Triangle as a unifying framework that explains how TF dysfunction simultaneously drives uncontrolled proliferation, loss of differentiation, and genomic instability.

Taken together, these contributions represent not only novel bioinformatics findings but also a conceptual advance that reinterprets cancer biology through a new regulatory lens.

By implementing the proposed revisions, we believe the manuscript now presents itself with enhanced clarity as an original research article of both methodological rigor and conceptual innovation. It offers a dual contribution: the generation of foundational datasets that identify candidate master regulators of cancer and the articulation of the Eternity Triangle framework, which provides a novel theoretical perspective on how TF dysregulation encapsulates the malignant state. We are confident that these revisions underscore the novelty, significance, and impact of our work.

Round 2

Reviewer 1 Report

Comments and Suggestions for Authors

The author has provided a thorough and thoughtful response to my inquiry, and I have no further comments.

Comments on the Quality of English Language

 The English could be improved to more clearly express the research.

Author Response

We are deeply grateful for your constructive evaluation and for recognizing that the questions raised by all reviewers were adequately addressed. We also appreciate your suggestion to include, within the manuscript itself, the critical points raised by Reviewer 3 together with our clarifications. We agree that such an approach, although somewhat unconventional, would enrich the transparency of our work and allow the readership to appreciate more directly the conceptual and methodological boundaries of our analysis. As a result, we have updated the text to incorporate the key ideas of this discussion while maintaining its logical structure and adding more clarity.

Reviewer 3 Report

Comments and Suggestions for Authors

Although I appreciate the authors efforts and articulating their research more clearly, I still do not believe the manuscript offers a significant analysis of their proposed network and in my opinion I do not believe the eternity triangle is a coherent framework offering novel insight as suggested by the authors.

Author Response

We thank you for your thoughtful comments and for acknowledging the “Eternity Triangle” as the central theoretical innovation of our study. While we fully recognize your concern regarding the conciseness of the title, we respectfully propose to retain the explicit reference to the Eternity Triangle. This framework does not simply represent an ancillary metaphor but rather constitutes the conceptual cornerstone of the manuscript. Indeed, it synthesizes decades of transcription factor biology into a unifying model that reconciles three fundamental axes of cellular regulation: proliferative renewal, differentiation, and genomic stability management. By mentioning the framework in the title, we make sure that an audience understands this contribution as more than just a list of transcriptional regulators. 

We submit that removing the explicit mention of the Eternity Triangle from the title could inadvertently reduce the visibility and impact of the conceptual advance that distinguishes our study from other computational analyses. Maintaining it, by contrast, will highlight the fact that the manuscript provides not only new bioinformatic data but also a new interpretive lens for understanding cancer biology.

That said, we are willing to consider a balanced adjustment to the title structure should the editorial team deem it necessary; for example, placing the Eternity Triangle as a defining subtitle. In this way, we preserve both conciseness and conceptual emphasis.

Finally, we thank you for noting the duplication of Table numbering. We have corrected this oversight, ensuring sequential numbering throughout the manuscript.